# EEG signature of breaks in embodiment in VR

**Thibault Porssut**[1,2,3,4☯], **Fumiaki Iwane**[4,5☯¤], **Ricardo Chavarriaga**[4,6], **Olaf Blanke**[2,7], **José del R. Millán**[4,5,8], **Ronan Boulic**[2‡*], **Bruno Herbelin**[1,2‡*]

**1** Immersive Interaction Research Group (IIG), Ecole Polytechnique Fédérale de Lausanne (EPFL), Lausanne, Switzerland, **2** Laboratory of Cognitive Neuroscience (LNCO), Ecole Polytechnique Fédérale de Lausanne (EPFL), Lausanne, Switzerland, **3** Altran Lab, Capgemini Engineering, Paris, France, **4** Ecole Polytechnique Fédérale de Lausanne (EPFL), Lausanne, Switzerland, **5** Department of Electrical and Computer Engineering, University of Texas at Austin, Austin, TX, United States of America, **6** Center for Artificial Intelligence, School of Engineering, Zurich University of Applied Sciences (ZHAW), Winterthur, Switzerland, **7** Department of Neurology, Geneva University Hospitals, Geneva, Switzerland, **8** Department of Neurology, University of Texas at Austin, Austin, TX, United States of America

☯ These authors contributed equally to this work.
¤ Current address: Human Cortical Physiology and Neurorehabilitation Section, NINDS, NIH, Bethesda, MA, United States of America
‡ RB and BH also contributed equally to this work.
* ronan.boulic@epfl.ch (RB); bruno.herbelin@epfl.ch (BH)

**Data Availability Statement:** The entire anonymized dataset is available on the zenodo.org platform under the DOI 10.5281/zenodo.7524989. The following link allows accessing it online: https://doi.org/10.5281/zenodo.7524989.

## Abstract

The brain mechanism of embodiment in a virtual body has grown a scientific interest recently, with a particular focus on providing optimal virtual reality (VR) experiences. Disruptions from an embodied state to a less- or non-embodied state, denominated Breaks in Embodiment (BiE), are however rarely studied despite their importance for designing interactions in VR. Here we use electroencephalography (EEG) to monitor the brain's reaction to a BiE, and investigate how this reaction depends on previous embodiment conditions. The experimental protocol consisted of two sequential steps; an induction step where participants were either embodied or non-embodied in an avatar, and a monitoring step where, in some cases, participants saw the avatar's hand move while their hand remained still. Our results show the occurrence of error-related potentials linked to observation of the BiE event in the monitoring step. Importantly, this EEG signature shows amplified potentials following the non-embodied condition, which is indicative of an accumulation of errors across steps. These results provide neurophysiological indications on how progressive disruptions impact the expectation of embodiment for a virtual body.

## Introduction

The integration of an avatar in virtual reality (VR) applications can evoke users' Sense of Embodiment (SoE) towards their virtual avatar. It is often proposed [1–5] that the SoE yields from the congruent association of the following three neurological processes related to the neuroscientific study of bodily self-consciousness; i) the sense of agency [6, 7], ii) the sense of body ownership [8–11] and iii) the sense of self-location [12–14]. Importantly, the subjective experience of embodiment is significantly altered if at least one of the three components is not

**Funding:** This work was supported by the Swiss National Science Foundation (project 'Immersive Embodied Interactions', 200020.178790), the Hasler Foundation (16070), and by the Swiss National Center of Competence in Research in Robotics (NCCR). The funders had no role in study design, data collection and analysis, decision to publish, or preparation of the manuscript.

**Competing interests:** The authors have declared that no competing interests exist.

respected [4, 5, 15]. It was further observed that violations of these conditions occurring after the successful induction of embodiment cause a disruption of SoE. Kokkinara et al. [16] denoted such disruptions as "Breaks", further identified as "Breaks in Embodiment" (BiE) by Porssut et al. [17] thereby providing a working definition for events interrupting embodiment and lowering the level of SoE to the point of impeding the immersive VR experience. These observations outline that the embodiment for a virtual body is, in essence, a simulation of the natural experience of embodying a real body [3, 18, 19]. As such, the subjective experience of embodiment for an avatar would be more an expectation that is satisfied than a new feeling that is induced. This view corroborates the observation that, once embodied in an avatar body, "people have some subjective and physiological responses as if it were their own body" [4].

Investigating BiE is key for understanding the cognitive mechanisms of virtual embodiment because, to the opposite of studies evaluating the overall subjective experience under given conditions and over relatively long periods of time, they are time-locked to a disembodiment event and less sensitive to cognitive biases. Importantly, research on BiE provide the appropriate conditions for the electrical neuroimaging study of their associated brain activity with electroencephalography (EEG), and can thus help in understanding if the building and disruption of embodiment are more continuous or discrete processes. Recent works [17, 20–23] observing the modulation of Error-related Potentials (ErrPs) induced by disruptions in VR already showed that the investigation of BiE can provide indirect assessments of embodiment. The neural mechanisms observed in these studies however overlapped with reactions that are not specific to embodiment, linked for instance to the violation of motor intentions [20–22], or did not allow concluding on the specificity of embodiment in absence of condition without error or of contrast with/without embodiment [23].

To circumvent former works' limitations and evaluate how virtual embodiment for an avatar is altered in a BiE event, we introduce a new task aiming at eliciting error perceptions specifically targeting a disruption of the SoE. In particular, we are interested in observing if the detection of a BiE depends on previous alterations in the embodiment conditions. Showing a modulation of the BiE would indeed reflect a successive accumulation of evidence contradicting the expectation for embodying an avatar. The absence of modulation would conversely depict BiEs as transient points of rupture in an established mental state. To test this hypothesis, we designed an experiment in which participant are asked, following an induction phase providing either reinforcement or violation of agency, to evaluate the presence of a potential disruption of body ownership. EEG was recorded in order to analyse the modulation of the ErrPs provoked by this disruption, and all factors are manipulated in a factorial design.

## Materials and methods

### Experimental protocol

19 healthy, right-handed subjects participated in the study (6 women, 22.9 ± 2.1 years (mean ± standard deviation (SD)). All participants had normal or corrected-to-normal vision and gave informed consent prior to their participation. The study was undertaken in accordance with the ethical standards as defined in the Declaration of Helsinki and was approved by the Ethical commission of Canton de Vaud on research involving human subjects (n°2018-01601).

During the experiment, participants wore a head mounted display with 1440 × 1440 resolution per eye, covering 110 degree of field of view at 90 Hz (The Explorer Headset, Lenovo). To eliminate auditory noise, we used a pair of in-ear headphones with active noise cancelling (QuietComfort 20, Bose) to play a non-localized white noise. Participants' full body motion

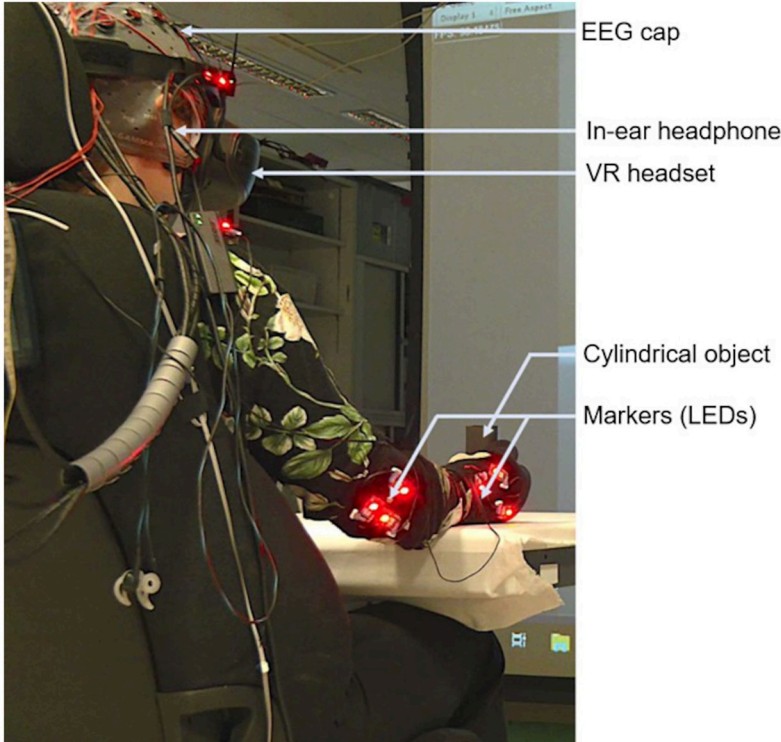

**Fig 1. Experimental setup.** Subjects were equipped with an EEG cap, a VR headset and headphones. A total of 17 markers (LEDs) were strapped on their body to track the head, shoulders, elbows and hands.

was captured by a motion capture system with 18 cameras and 17 markers on their body (ImpulseX2, PhaseSpace). Fig 1 shows the experimental setup.

Participants immersed in a 3D environment saw their avatar in first person view. The virtual chair, table and 3D avatar were calibrated to match the position, orientation and dimensions of the actual environment. Both the participant and the avatar held a cylindrical object in their right hand, preventing participants from moving their fingers while maintaining a visuo-proprioceptive and visuo-tactile coherence. The VR environment was implemented with Unity 3D (2019.2.0f1). Participants' movements were reproduced through animation of the virtual avatar using FinalIK (https://root-motion.com).

The experiment took place in different phases as follows (Fig 2A). After calibration and training, the experimental blocks started with a reaching task phase of 6s, with the aim to establish a baseline giving enough time for participants to make the experience of a high level of embodiment towards the virtual avatar. During this phase, participants were instructed to perform reaching movements to four different targets.

The experimental task itself consisted of two steps (Fig 2B). Each trial started with an active induction step during which participants were instructed to turn the hand twice (wrist rotation). In the following monitoring step, a fixation cross appeared above the hand, and participants should remain still and fix their gaze on the cross. The succession of the two steps is key for our observations; first we induce a sense of embodiment (or not) for the avatar and second, we disrupt this sense of embodiment (or not) while participants are passively monitoring the virtual arm (to prevent any eye movement or motion artifacts in the EEG signals). The experimental conditions thus correspond to the 2 × 2 conditional matrix affecting the induction

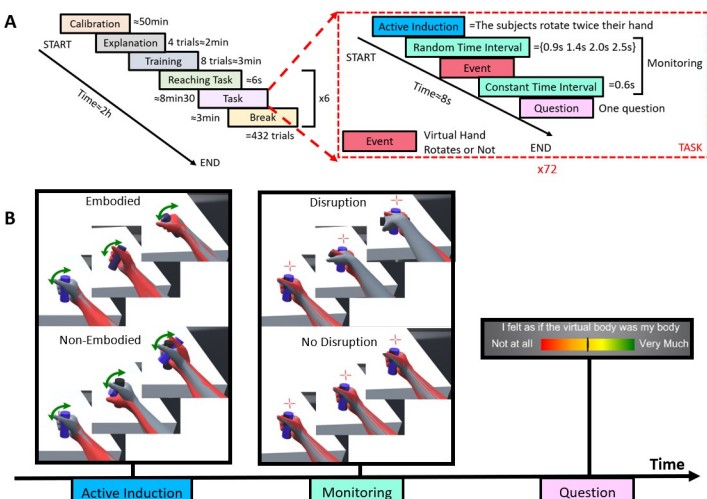

**Fig 2. Illustration of the experiment design. A:**The experiment consisted of six phases; i) calibration, ii) explanation, iii) training, iv) reaching task, v) experimental task and some breaks. The explanation phase consisted of four trials to illustrate each experimental condition. Subjects performed eight more trials during the training phase to ensure that they understood and performed the task correctly. **B:** The experimental task is the succession of 2 steps. During the active induction step, two conditions can be presented to the subject, either *Embodied* or *Non-Embodied* with the avatar. In the monitoring step, two conditions can occur to either produce a *Disruption* (the right virtual hand rotates by itself) or *No Disruption* (nothing happens). In the figures, the grey arm is avatar's arm movement (as seen by the subject in VR) and the red arm corresponds to participants' physical arm movement (not shown in VR, shown here for illustration purposes). Following the 2 steps, participant's subjective rating of body ownership is gathered.

step, *Embodied* or *Non-Embodied*, and the monitoring step, Disruption or No Disruption. In the Embodied condition, the virtual hand followed the participant's hand doing the wrist rotation. Conversely, the virtual hand remained still in the *Non-Embodied* condition. The two conditions of the monitoring step occurred after a randomized fixation time (randomly picked from 0.9 s, 1.5 s and 2.1 s). These randomized time intervals were to prevent participants from knowing the exact starting time of the next step. In the No Disruption condition, the virtual hand and the physical hand of the participant remained still. In the Disruption condition, the virtual hand autonomously performed a wrist rotation while the physical hand remained still. At the end of the monitoring step (0.6 s after the event if it occurred), participants were prompted to answer a questionnaire (see in the next section Subjective Rating). We employed the different task between step 1 and step 2 to remove active motor action of their wrist and to temporally align when participants perceive Break-in-Embodiment. Due to time constraints, no questions were asked at the end of step 1. We ensured that participants were embodied or not during this first step thanks to the answers during *Embodied/No Disruption* and *Non-Embodied/No Disruption condition*.

Participants performed 360 trials of the experimental task, distributed in 5 blocks of 72 trials. The number of trials in *Embodied* and *Non-Embodied* conditions was identical (50%). Because the successful elicitation of ErrPs depends on the unexpected nature of the occurrence of an event, the experiment presented more often trials without disruption than with. The ratio of 33% of Disruption trials for the monitoring step was determined based on previous studies [18, 19, 22–24]. In each block, we ensured the ratio of each condition and the trials were counterbalanced to compensate for the order effect. Thus, each participant performed 120 trials of *Non-Embodied/No Disruption* and *Embodied/No Disruption*, 60 trials of *Non-Embodied/Disruption* and *Embodied/Disruption* The number of trials was not available to

participants. Participants took a break for at least three minutes after each block, starting with the reaching task again before resuming the experimental tasks.

## Subjective rating

As in previous studies [17, 24] subjects were asked to rate their agreement to the affirmation "I felt as if the virtual body was my body" on a visual color gradient scale (see Fig 2B) ranging from "No at all (red, on the left) to Very Much (green, to the right). This measure of body ownership was adapted from previous work [16, 25]. Subjects were asked to use the full scale by moving horizontally a cursor with the head and to validate with a trigger button in their left hand. Although no numerical feedback was provided to subjects, answers were recorded in a continuous scale from 0 to 100. Because the experimental design requests a large number of repetitions, we wanted to minimize the number of questionnaire items. We decided not to ask for a subjective rating of the sense of agency as it would only have been useful to confirm that participants were aware of our manipulation, and we had no reason to hypothesize that it would not be the case [17].

As we observed that data were not normally distributed using one-sample KolmogorovSmirnov test, we performed two-way repeated measures Friedman ANOVA to investigate the main effect of embodiment (*Embodied* vs *Non-Embodied*) and disruption (*Disruption* and *No Disruption*). To investigate the interaction effects, Wilcoxon signed-rank test was applied for each pair of the conditions. The false discovery rate (FDR) was corrected for the six withingroup comparison using the Benjamini-Hochberg procedure. The effect size is computed using the scaled robust Cohen's standardized mean difference (dr) for non-normal residuals [26, 27].

## EEG signal processing

EEG signals were recorded throughout the experiment using three synchronized g.USBAmp (g.tec medical technologies, Austria) and 32 active electrodes located following 10/10 international system [28]. EOG signal was simultaneously recorded with 3 active electrodes from above the nasion and below the outer canthi of the eyes. The ground electrode was placed on the forehead (AFz) and the reference electrode on the left earlobe. EEG and EOG signals were recorded at 512 Hz.

A channel rejection and subsequent spherical interpolation was performed on the EEG signals based on established methods [29, 30]. This process removed 1 ± 1 channels per participant. Subsequently, Independent Component Analysis (ICA) was performed after the high-pass filter (Noncausal 2nd order Butterworth filter with 1 Hz cutoff frequency) to remove artifactual independent components correlated with at least one of EOG signals. This process removed 2 ± 2 ICs. After artifactual ICs removal, signals were projected back to the channel space, and low-pass filtered at 30 Hz [31].

Processed EEG signals were then segmented into epochs within a time window of [-0.2, 0.6] s with respect to the onset of monitoring step. We restricted the subsequent EEG analysis to the monitoring step as participants performed motor actions in the induction step, which may be a confounding factor when evaluating humans' cognitive process. Contaminated EEG epochs were identified and rejected based on the probability of occurrence [29, 30]. Three participants were removed from the subsequent analysis due to the limited number of clean EEG epochs; i.e., only less than 50% of epochs were kept after epoch rejection. For remaining participants, this process removed 3.2 ± 1.6% of trials on average.

In order to identify a specific time window in which EEG was significantly modulated between the conditions, we used a non-parametric cluster-based permutation test [32].

Specifically, we applied this test to the temporal signals at Pz, Cz and FCz channels as these electrodes have been identified to provide maximal modulation induced by BiE [20–23, 33, 34]. Each sample underwent a one-way repeated measures ANOVA with 4 conditions (*Embodied/Disruption (ED)*, *Non-Embodied/Disruption (NED)*, *Embodied/No Disruption (END)*, *Non-Embodied/No Disruption(NEND)*), from which significant bins ($\alpha$ = 0.01) were further clustered. We applied cluster correction to keep the cluster which is significantly larger than the regions identified during the permutation test ($\alpha$ = 0.01). For post-hoc analyses, we computed the averaged amplitude at each identified time window and performed paired Student's t-tests to compare the mean amplitude within the clusters. Permutation tests were performed to estimate the significance of the results.

## Results

### Subjective ratings

Statistical analysis revealed a significant main effects of both embodiment ($X_2$ = 45.35, p < 0.0001) and disruption factors ($X_2$ = 9.53, p < 0.01) on the subjective ratings of embodiment. The *Embodied* condition (73.1 ± 19.9) yielded a higher sense of embodiment than the *Non-Embodied* condition (16.8 ± 17.4), and the No Disruption condition (51.5 ± 35.9) to higher scores than the Disruption condition (38.4 ± 30.3). As shown in Fig 3, we observed significant interaction effects for all pairwise comparisons (p < 0.01), except for the comparison between *Non-Embodied/Disruption* and *Non-Embodied/No Disruption* (p = 0.15).

### Electrophysiological measures

Fig 4B shows grand averaged signals at the Pz, Cz and FCz electrodes for each condition with respect to the onset of disruption events (t = 0) and in gray the time windows showing significant differences between conditions. Three different time windows have been identified, each of them corresponding to different event-related potentials. The first window corresponds to an error-related negativity [35] (ERN, [116–192] ms) at Pz. The second window corresponds to an error-positivity [36] (Pe, [218–294] ms) at Pz, Cz and FCz. Finally, the third window corresponds to a N400 wave [35] (N400 [447–600] ms) at Cz and FCz. The presence of these components have been reported in the previous ErrP studies in VR setup [20, 21]. Topographical

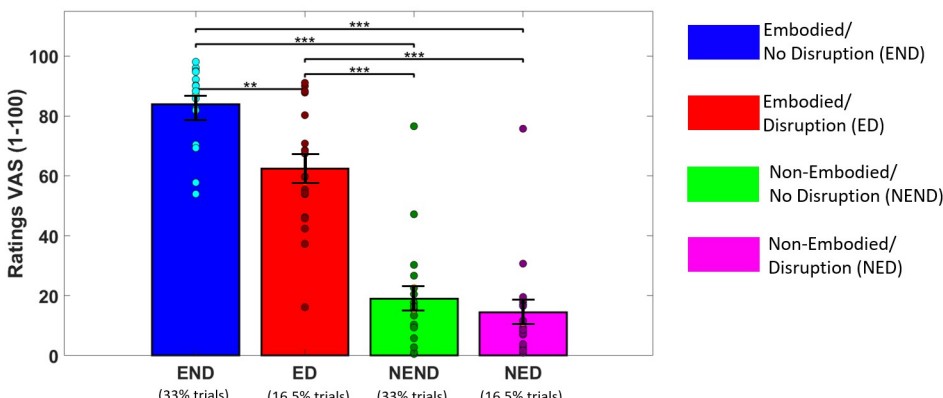

**Fig 3. Embodiment questionnaire results for all conditions.** Participants rated their sense of embodiment by answering to "*I felt as if the virtual body was my body*" after each trial in a visual analog scale (VAS) ranging from "Not at all" (0) to "Very Much" (100). Each dot represents an individual participant.*$Q_{FDR}$ < 0.05, **$Q_{FDR}$ < 0.01, ***$Q_{FDR}$ < 0.001.

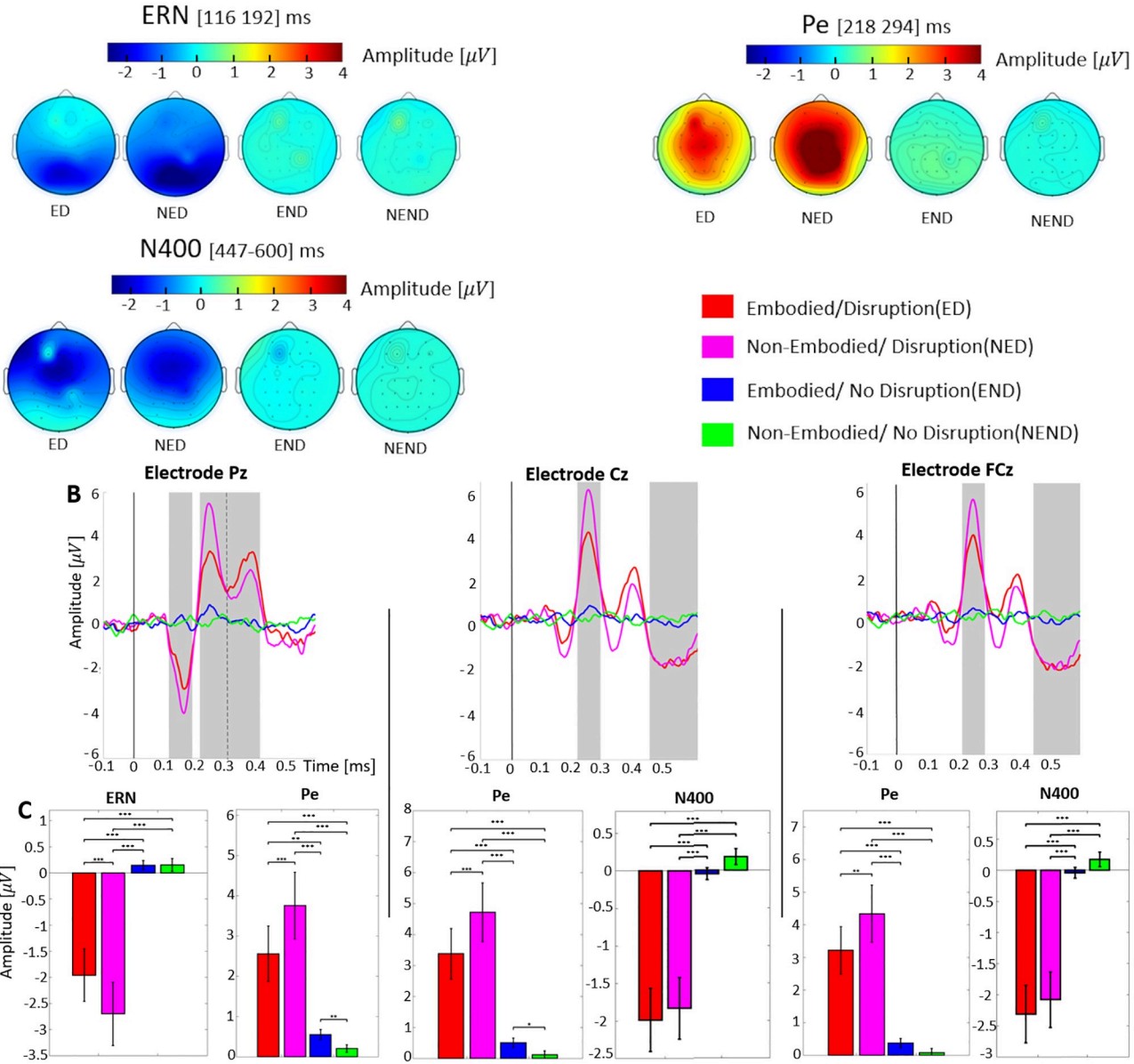

**Fig 4. ErrPs elicited by BiE at event onset (t = 0).** Three components were identified: ERN is defined as the first negative deflection occurring between 100 and 200ms, Pe is the first positive deflection occurring between 200 and 300 ms and N400 is the negative deflection occurring between 400 and 600ms. **A:** Topographical distribution in the four conditions during the time window of the significant effect for ERN, Pe and N400. **B:** Time-locked average time-courses at parietal (Pz), central (Cz), and front-central (FCz) electrodes. The vertical black line corresponds to t = 0s when the hand animation starts. The gray background areas represent the time windows in which two-way ANOVA showed a main effect between conditions($Q_{FDR} < 0.01$). **C:** Mean amplitudes at each time window identified (in gray) $^*Q_{FDR} < 0.05$, $^{**}Q_{FDR} < 0.01$, $^{***}Q_{FDR} < 0.001$.

representation of the *Disruption condition* (Fig 4A) revealed the focal activation of the parietal, central and frontal area for ERN, Pe and N400, respectively. The event related potentials of the *No Disruption* condition did not exhibit any prominent electrophysiological deflections in all the three channels. We did not observe any effect in the EEG signals associated with different interval durations.

Regarding ERN, the two-way repeated measures Friedman ANOVA used to investigate the main effect of the embodiment condition (*Embodied* or *Non-Embodied*) and the disruption

**Table 1. Results of post-hoc analysis for ERN at the electrode Pz, Cz and FCz.**

| ERN | NED/ED | NED/END | NED/NEND | ED/END | ED/NEND | END/NEND |
|---|---|---|---|---|---|---|
| Pz | $t(18) = 3.3$<br>$p < 0.001$<br>$Q_{FDR} < 0.001$<br>$dr = 0.48$ | $t(18) = -4.35$<br>$p < 0.001$<br>$Q_{FDR} < 0.001$<br>$dr = 1.08$ | $t(18) = -4.51$<br>$p < 0.001$<br>$Q_{FDR} < 0.001$<br>$dr = 1.03$ | $t(18) = -3.88$<br>$p < 0.001$<br>$Q_{FDR} < 0.001$<br>$dr = 1.23$ | $t(18) = -3.88$<br>$p < 0.001$<br>$Q_{FDR} < 0.001$<br>$dr = 1.13$ | $t(18) = -0.03$<br>$p = 0.49$<br>$Q_{FDR} = 0.49$<br>$dr = 0.058$ |
| Cz | n/a | n/a | n/a | n/a | n/a | n/a |
| FCz | n/a | n/a | n/a | n/a | n/a | n/a |

condition (*Disruption* or *No Disruption*) revealed a significant main effect on the disruption condition (p < 0.0001) but not on the embodiment condition (p = 0.37). Subsequent post-hoc analysis showed significant differences in ERN amplitude between all the possible pairs, except for *Embodied/No Disruption* and *Non-Embodied/No Disruption* (see Table 1).

Regarding Pe, we observed the main effect of the disruption condition for all the three channels (p < 0.001 for Pz and p < 0.0001 for Cz and FCz). However, we did not observe any main effect of the embodiment condition (p = 0.38 for Pz, p = 0.47 for Cz and p = 0.52 for FCz). Subsequent post-hoc analyses showed that the Pe amplitude was significantly different between all possible pairs; except a pair of *Embodied/No Disruption* and *Non-Embodied/No Disruption* at FCz channel (see Fig 4C and Table 2).

Regarding N400, as for the ERN and Pe, statistical analysis revealed a significant main effect of the disruption condition (p < 0.0001 for Cz and FCz). However, we did not observe any main effect of the embodiment condition (p = 0.11 for Cz and p = 0.14 for FCz). Subsequent post-hoc analyses showed that N400 amplitude was significantly different between *Non-Embodied/Disruption* against *Embodied/No Disruption*, *Non-Embodied/Disruption* against *Non-Embodied/No Disruption*, *Embodied/Disruption* against *Embodied/No Disruption* and *Embodied/Disruption* against *Non-Embodied/No Disruption* for both Cz and FCz channels (see Fig 4C and Table 3).

## Discussion

Our results show that both manipulations, during a first induction step and in a subsequent disruption step, affected the subjective ratings of body ownership. First, we observe a strong and significant effect of the manipulation of agency done in the first experimental step (*Embodied* vs. *Non-Embodied* conditions). Second, we also confirm that the visuo-proprioceptive disruption occurring in the second experimental step (with or without disruption) was

**Table 2. Results of the post-hoc analysis for Pe at the electrode Pz, Cz and FCz.**

| Pe | NED/ED | NED/END | NED/NEND | ED/END | ED/NEND | END/NEND |
|---|---|---|---|---|---|---|
| Pz | $t(18) = -3.73$<br>$p < 0.001$<br>$Q_{FDR} < 0.001$<br>$dr = 0.084$ | $t(18) = 3.93$<br>$p < 0.001$<br>$Q_{FDR} < 0.001$<br>$dr = 1.03$ | $t(18) = 4.26$<br>$p < 0.001$<br>$Q_{FDR} < 0.001$<br>$dr = 1.18$ | $t(18) = 3.01,$<br>$p < 0.01$<br>$Q_{FDR} < 0.01$<br>$dr = 0.5$ | $t(18) = 3.44$<br>$p < 0.001$<br>$Q_{FDR} < 0.001$<br>$dr = 0.58$ | $t(18) = 2.65$<br>$p < 0.01$<br>$Q_{FDR} < 0.01$<br>$dr = 0.98$ |
| Cz | $t(18) = -3.48$<br>$p < 0.001$<br>$Q_{FDR} < 0.001$<br>$dr = 0.69$ | $t(18) = 4.43$<br>$p < 0.001$<br>$Q_{FDR} < 0.001$<br>$dr = 0.91$ | $t(18) = 4.71$<br>$p < 0.001$<br>$Q_{FDR} < 0.001$<br>$dr = 0.99$ | $t(18) = 3.59$<br>$p < 0.001$<br>$Q_{FDR} < 0.001$<br>$dr = 0.99$ | $t(18) = 3.87,$<br>$p < 0.001$<br>$Q_{FDR} < 0.001$<br>$dr = 1.1$ | $t(18) = 1.86$<br>$p < 0.05$<br>$Q_{FDR} < 0.05$<br>$dr = 0.38$ |
| FCz | $t(18) = -2.74$<br>$p < 0.01$<br>$Q_{FDR} < 0.01$<br>$dr = 0.55$ | $t(18) = 4.61$<br>$p < 0.001$<br>$Q_{FDR} < 0.001$<br>$dr = 0.95$ | $t(18) = 4.71$<br>$p < 0.001$<br>$Q_{FDR} < 0.001$<br>$dr = 1.02$ | $t(18) = 4.1$<br>$p < 0.001$<br>$Q_{FDR} < 0.001$<br>$dr = 1.00$ | $t(18) = 4.23$<br>$p < 0.001$<br>$Q_{FDR} < 0.001$<br>$dr = 1.24$ | $t(18) = 1.55$<br>$p = 0.06$<br>$Q_{FDR} = 0.06$<br>$dr = 0.61$ |

Table 3. Results of the post-hoc analysis for N400 at the electrode Cz and FCz.

| N400 | NED/ED | NED/END | NED/NEND | ED/END | ED/NEND | END/NEND |
|---|---|---|---|---|---|---|
| Pz | n/a | n/a | n/a | n/a | n/a | n/a |
| Cz | $t(18) = -0.59$<br>$p = 0.28$<br>$Q_{FDR} = 0.28$<br>$dr = 0.084$ | $t(18) = -3.88$<br>$p < 0.001$<br>$Q_{FDR} < 0.001$<br>$dr = 1.13$ | $t(18) = -4.32$<br>$p < 0.001$<br>$Q_{FDR} < 0.001$<br>$dr = 1.16$ | $t(18) = -4.04$<br>$p < 0.001$<br>$Q_{FDR} < 0.001$<br>$dr = 1.02$ | $t(18) = -4.86$<br>$p < 0.00$<br>$Q_{FDR} < 0.001$<br>$dr = 1.11$ | $t(18) = -1.4$<br>$p = 0.08$<br>$Q_{FDR} = 0.096$<br>$dr = 0.79$ |
| FCz | $t(18) = -0.73$<br>$p = 0.23$<br>$Q_{FDR} = 0.23$<br>$dr = 0.038$ | $t(18) = -4.28$<br>$p < 0.001$<br>$Q_{FDR} < 0.001$<br>$dr = 1.15$ | $t(18) = -4.46$<br>$p < 0.001$<br>$Q_{FDR} < 0.001$<br>$dr = 1.15$ | $t(18) = -4.35$<br>$p < 0.001$<br>$Q_{FDR} < 0.001$<br>$dr = 1.28$ | $t(18) = -5.11$<br>$p < 0.001$<br>$Q_{FDR} < 0.001$<br>$dr = 1.32$ | $t(18) = -1.13$<br>$p = 0.13$<br>$Q_{FDR} = 0.16$<br>$dr = 0.61$ |

clearly noticed and led to significantly different ratings in the *Embodied* condition. Third and most interestingly, this difference is not observed in the *Non-Embodied* condition, thus showing the influence of the first step on the subsequent manipulation.

The electrophysiological signature of ErrPs shows a successful induction of ERN and Pe following our experimental disruption. As expected, no ErrPs were elicited in absence of visuo-proprioceptive conflict [37]. More importantly, the ErrPs components were modulated by the prior step of manipulation of the sense of agency. We observe an increased level of amplitude of ERN and Pe for the *Non-Embodied* condition compared to the *Embodied* condition. These results corroborate the principle of the accumulation of errors revealed by Steinhauser et al. [38] and the work of Chang et al. [39, 40] who provided neurophysiological evidence of post-error adjustment by showing amplified ErrPs in trials following errors. In line with this observation, our data reveal a mechanism of error monitoring, as for post-error adjustments, accumulating evidence against the expected state and leading to an eventual disruption (i.e. possible BiE). Extending these observations, our results demonstrate that accumulation of errors can occur when different types of cognitive process were disrupted, i.e. the sense of agency and the sense of body ownership, suggesting that these different factors accumulate into a global error of the experience of embodiment. This can be further corroborated with the mutual influence between manipulations observed with subjective ratings.

Furthermore and extending the ERN and Pe results, we observe N400 in the front-central area of the brain. This potential was originally identified when participants perceived semantic errors during linguistic processing [41], or in a performance monitoring tasks upon the perception of the erroneous actions [42]. Modulations of the N400 were also observed in VR when subjects were prevented from achieving their movement while embodied in a virtual avatar [20, 21]. In our study, the amplitude of N400 is the same regardless of the condition in the induction step, inline with results of previous similar works [22].

## Conclusion

The present study investigated the participants' subjective ratings of embodiment and their electrical neuroimaging data in order to reveal the mechanism of breaks in embodiment. In line with previous experiments [23, 43, 44] our manipulation successfully elicited ErrPs when participants experienced a conflict between their real and virtual bodies (visuo-proprioceptive disruption). Extending these previous works, our results show that the level of SoE and the amplitude of ErrPs are modulated by prior conditions manipulating the sense of embodiment. Specifically, we observe significantly lower subjective ratings of body ownership and larger amplitudes of ERN and Pe components when participants previously had an unfavorable experience of embodiment (no agency for the avatar hand), as compared to cases following a

positive embodiment induction (visuo-motor synchrony with avatar hand). These results thus tend to show that an accumulation of evidence against the expectation of embodiment leads to higher reactions to what are otherwise identical disruptions. Importantly, we demonstrate with our two-steps manipulation that different conflicts of embodiment (visuo-motor and visuo-proprioceptive) can combine into what could be a unified error monitoring mechanisms of embodiment, providing one of the first evidence of a neural response to a BiE.

These observations give some insight on the neural mechanisms of virtual embodiment. If the embodiment for a virtual body was conflicting with the embodiment for the real one, a VR experience would start with a low a priori embodiment for the avatar, that builds up only if conditions are favorable to confirm that the virtual body is the one owned, and that abruptly breaks upon any contradicting evidence. What we rather observe is an accumulation of contradicting evidences against the expectation of embodiment for the avatar. Our experimental design however does not allow observing the neural response to each successive disruptions. This is mostly due to difficulties in performing a clean ErrPs analysis at times when participants are performing motor actions (artefacts from movement of participants, widespread brain activity during motor execution), but would be worth investigating with different disruptions and more advanced EEG analysis. Going further with the combination of conditions of disruptions in reality and in mixed reality would even allow investigating the specific nature of a break in virtual embodiment, eventually even allowing to determine if the expectation of embodiment for the virtual body is due to an expectation of continuity from the embodiment for the real body.

As an outlook, and since ErrPs are not subject to the same degree of introspection as the standard presence questionnaires and can be done without interrupting for explicit user feedback, they could be used to implicitly detect disruptions of embodiment. This would allow conducting background evaluations of the subject's immersive experience, and be used for quality assessment of VR systems and paradigms. This approach could also be beneficial for the clinical evaluation of embodiment of prosthetic limbs, with benefits for reducing phantom limb distortions and pain in hand [45] or leg [46] amputee.

## Author Contributions

**Conceptualization:** Thibault Porssut, Fumiaki Iwane, Ricardo Chavarriaga, Bruno Herbelin.

**Data curation:** Thibault Porssut, Fumiaki Iwane.

**Formal analysis:** Thibault Porssut, Fumiaki Iwane.

**Funding acquisition:** Olaf Blanke, José del R. Millán, Ronan Boulic, Bruno Herbelin.

**Investigation:** Fumiaki Iwane.

**Methodology:** Thibault Porssut, Fumiaki Iwane, Ricardo Chavarriaga, Ronan Boulic, Bruno Herbelin.

**Software:** Thibault Porssut, Fumiaki Iwane.

**Supervision:** Ricardo Chavarriaga, Olaf Blanke, José del R. Millán, Ronan Boulic, Bruno Herbelin.

**Validation:** Thibault Porssut, Fumiaki Iwane, Bruno Herbelin.

**Visualization:** Thibault Porssut, Fumiaki Iwane.

**Writing – original draft:** Thibault Porssut, Fumiaki Iwane.

**Writing – review & editing:** Thibault Porssut, Fumiaki Iwane, Ricardo Chavarriaga, Olaf Blanke, José del R. Millán, Ronan Boulic, Bruno Herbelin.

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
