## [Decision Letter · Decision Letter 0]

13 Dec 2022

PONE-D-22-23485EEG signature of breaks in embodiment in VRPLOS ONE

Dear Dr. Herbelin,

Thank you for submitting your manuscript to PLOS ONE. After careful consideration, we feel that it has merit but does not fully meet PLOS ONE’s publication criteria as it currently stands. Therefore, we invite you to submit a revised version of the manuscript that addresses the points raised during the review process.

We look forward to receiving your revised manuscript.

Kind regards,

Jane Elizabeth Aspell, PhD

Academic Editor

PLOS ONE

Journal Requirements:

2. Please change "female” or "male" to "woman” or "man" as appropriate, when used as a noun (see for instance https://apastyle.apa.org/style-grammar-guidelines/bias-free-language/gender).

This work was supported by the Swiss National Science Foundation (project ‘Immersive Embodied Interactions’, 200020.178790), the Hasler Foundation, Switzerland, and by the Swiss National Center of Competence in Research in Robotics (NCCR).

Reviewers' comments:

Reviewer's Responses to Questions

**Comments to the Author**

1. Is the manuscript technically sound, and do the data support the conclusions?

Reviewer #1: Yes

Reviewer #2: Yes

2. Has the statistical analysis been performed appropriately and rigorously? 

Reviewer #1: Yes

Reviewer #2: Yes

3. Have the authors made all data underlying the findings in their manuscript fully available?

Reviewer #1: Yes

Reviewer #2: Yes

4. Is the manuscript presented in an intelligible fashion and written in standard English?

Reviewer #1: Yes

Reviewer #2: Yes

5. Review Comments to the Author

Reviewer #1: The authors present an interesting study on embodiment in virtual reality studying the neural mechanisms during disruption of ownership illusions, otherwise known as breaks in embodiment, through electrical neuroimaging data acquisition. In line with previous work, the authors report error-related potentials when visuo-proprioceptive correlations are not present (disruption of agency) and that the amplitude of the errors is modulated under the different experimental condition of manipulation of the sense of embodiment.

In general, the study is well-structured and presented regarding the introduction, methodology, data analysis, and discussion. The authors report adequately on the background literature to later support their hypothesis and argue their findings. I find the study of great importance for researchers in the filed and for advancing the knowledge.

I do have some questions on a couple of points that were not clear not.

First, regarding the subjective reports, why was only one question included in the assessment referring to the extent participants felt that the virtual body was theirs (‘I felt as if the virtual body was my body’), whereas in previous studies additional questions also related to agency are usually included? See examples Peck TC and Gonzalez-Franco M (2021) Avatar Embodiment. A Standardized Questionnaire or Roth D and Latoschik ME (2020) among others.

Second, though the number of participants is relatively low the within-groups design allows for good comparisons. Nonetheless, I did not see in the text if the different experimental conditions were counterbalance and now these were presented to participants over the different blocks and trials.

Last, the figures in the submitted manuscript are of low quality and need to be upgraded, and a video demonstration of the experimental setup would help readers better understand the scenario and discussed manipulations.

Reviewer #2: The article deals with an interesting topic, namely the breaks in the embodiment by providing behavioral and electroencephalography original data. Overall, the results are intelligible and well presented, including the clear mention of the limits (e.g. absence of clean error-related potentials analysis during motor actions), which are then manifestly available to the reader. I provide below just a few point-by-point suggestions and comments that I hope will contribute to improving the quality of the authors’ work.

- Given the relevance of the manipulation, I suggest the authors clearly state and repeat in the initial part of the method section the usefulness of each of the two steps included in the experimental phase (i.e. embodiment to create the embodiment and disruption to manipulate the feeling eventually created in the previous phase). Personally, I do not find the term 'monitoring' entirely appropriate and would suggest finding an alternative formulation

- Wisely the authors include different time intervals within the experimental phase, however, this experimental choice is not fully explained. I suggest adding an explanation of the choice

- In principle, the same task as in step 1 could be repeated to disrupt the embodiment. Can the authors specify the rationale behind the choice of changing the task from a visuo-motor to a visuo-proprioceptive (e.g. avoiding artifacts?) and discuss the possible consequences or whether or not they believe there may be any?

- the authors only investigate embodiment at the end of the entire experimental procedure (steps 1 and 2). Why was embodiment not also investigated after step 1 alone, to verify (at least on a behavioral level) that the manipulation was successful?

- Does the duration of the embodiment phase (time interval from active induction to monitoring phase) have an impact on the results? are there differences related to the different time intervals involved?

- In general, I find the figures helpful and well thought out. In figure 1, adding labels to explain the different components of the setup could improve the quality of the picture

- Recent studies have demonstrated the relevance of the virtual reality environment (coupled with sensory stimulation) in the field of rehabilitation (Risso Preatoni et al. 2022: https://doi.org/10.1016/j.isci.2022.104129; Risso & Valle https://doi.org/10.1007/s40141-022-00350-x; Serino et al. 2022 https://www.cairn.info/revue-de-neuropsychologie-2022-1-page-15.html ; Rognini et al 2019 http://dx.doi.org/10.1136/jnnp-2018-318570 ). It would be interesting to discuss the potential impact of these findings on the clinical setting.

Typos:

- Section 1 introduction : […]Errorrelated Potentials (ErrPs) induced by disruptions in VR already […]

- Section 2.2. Subjective rating: […] withingroup comparison using the Benjamini-Hochberg[…]

6. PLOS authors have the option to publish the peer review history of their article (what does this mean?). If published, this will include your full peer review and any attached files.

Reviewer #1: No

Reviewer #2: No

---

## [Author Response · Author response to Decision Letter 0]

20 Jan 2023

Dear Editor and Reviewers,

Thank you for reviewing our manuscript entitled “EEG signature of breaks in embodiment in VR” and for providing constructive feedback. We greatly appreciate your comments and have studied them carefully. In the following, we address the reviewers’ comments point by point and explain how the paper has been revised accordingly.

For your convenience, a PDF response document is also available.

Yours sincerely,

Authors of the manuscript

Thibault Porssut, Fumiaki Iwane, Ricardo Chavarriaga, Olaf Blanke, José del R. Millán, Ronan Boulic and Bruno Herbelin

Reviewer #1: The authors present an interesting study on embodiment in virtual reality studying the neural mechanisms during disruption of ownership illusions, otherwise known as breaks in embodiment, through electrical neuroimaging data acquisition. In line with previous work, the authors report error-related potentials when visuo-proprioceptive correlations are not present (disruption of agency) and that the amplitude of the errors is modulated under the different experimental conditions of manipulation of the sense of embodiment.

In general, the study is well-structured and presented regarding the introduction, methodology, data analysis, and discussion. The authors report adequately on the background literature to later support their hypothesis and argue their findings. I find the study of great importance for researchers in the field and for advancing knowledge.

I do have some questions on a couple of points that were not clear not.

1. First, regarding the subjective reports, why was only one question included in the assessment referring to the extent participants felt that the virtual body was theirs (‘I felt as if the virtual body was my body’), whereas in previous studies additional questions also related to agency are usually included? See examples Peck TC and Gonzalez-Franco M (2021) Avatar Embodiment. A Standardized Questionnaire or Roth D and Latoschik ME (2020) among others.

Thank you for expressing this comment, which was indeed a design choice that might not appear obvious or justified. We added the following sentences to the manuscript;

In page 5, lines 119: “Because the experimental design requests a large number of repetitions, we wanted to minimize the number of questionnaire items. We decided not to ask for a subjective rating of the sense of agency as it would only have been useful to confirm that participants were aware of our manipulation, and we had no reason to hypothesize that it would not be the case17. “

2. Second, though the number of participants is relatively low the within-groups design allows for good comparisons. Nonetheless, I did not see in the text if the different experimental conditions were counterbalanced and now these were presented to participants over the different blocks and trials.

Thank you for noticing this missing information. We added the following sentences to the manuscript. 

In page 5, line 104: “In each block, we ensured the ratio of each condition and the trials were counterbalanced to compensate for the order effect. Each participant performed 120 trials of Non-Embodied/No Disruption and Embodied/No Disruption, 60 trials of Non-Embodied/Disruption and Embodied/Disruption. The number of trials was not available to participants. “

3. Last, the figures in the submitted manuscript are of low quality and need to be upgraded, and a video demonstration of the experimental setup would help readers better understand the scenario and discuss manipulations.

Sorry that the figure quality was lost in the pre-print to PLOS-ONE process: we will ensure that high quality figures are published. Please find a link to a video of the experiment:

https://youtu.be/Pg1cqpwDas8

Reviewer #2: The article deals with an interesting topic, namely the breaks in the embodiment by providing behavioral and electroencephalography original data. Overall, the results are intelligible and well presented, including the clear mention of the limits (e.g. absence of clean error-related potentials analysis during motor actions), which are then manifestly available to the reader. I provide below just a few point-by-point suggestions and comments that I hope will contribute to improving the quality of the authors’ work.

1. Given the relevance of the manipulation, I suggest the authors clearly state and repeat in the initial part of the method section the usefulness of each of the two steps included in the experimental phase (i.e. embodiment to create the embodiment and disruption to manipulate the feeling eventually created in the previous phase). Personally, I do not find the term 'monitoring' entirely appropriate and would suggest finding an alternative formulation

Thanks for your suggestion on how to improve the explanation of our experimental approach. We added the following sentence in the beginning of the paragraph explaining the two steps:

In page 3, line 77: “The succession of the two steps is key for our observations; first we induce a sense of embodiment (or not) for the avatar and second, we disrupt this sense of embodiment (or not) while participants are passively monitoring the virtual arm to prevent any eye movement or motion artifacts in the EEG signal).”

Regarding the choice of the term ‘monitoring’, it is based on the literature on ‘error monitoring’ and ‘performance monitoring’; while we understand that it might not be ideal, we couldn’t converge to a better alternative (e.g. ‘observation’ or ‘waiting’ are too passive, ‘task’ is incorrect as the task includes two steps)

2. Wisely the authors include different time intervals within the experimental phase, however, this experimental choice is not fully explained. I suggest adding an explanation of the choice

As pointed out, we employed the time intervals of 0.9, 1.5 and 2.1 s. The choice of these duration was to prevent participants from knowing the exact moment when the next step starts. We added the following sentence to the manuscript to describe our choice of the time intervals. 

In page 4, line 87: “These randomized time intervals were to prevent participants from knowing the exact starting time of the next step.”

3. In principle, the same task as in step 1 could be repeated to disrupt the embodiment. Can the authors specify the rationale behind the choice of changing the task from a visuo-motor to a visuo-proprioceptive (e.g. avoiding artifacts?) and discuss the possible consequences or whether or not they believe there may be any?

We modified the tasks between step 1 and step 2 for two reasons; to remove active motor actions and to temporally align when participants perceive Break-in-Embodiment. 

In page 4, line 93: “We employed different tasks between step 1 and step 2 to remove active motor action of their wrist and to temporally align when participants perceive Break-in-Embodiment.” 

4. The authors only investigate embodiment at the end of the entire experimental procedure (steps 1 and 2). Why was embodiment not also investigated after step 1 alone, to verify (at least on a behavioral level) that the manipulation was successful?

The experiment was already long (2h in total). So we had to restrict the number of questions. Moreover, thanks to the answers during the Embodied/No Disruption condition where participants were embodied with no disruption we could ensure that participants were embodied during the first step. On the contrary, thanks to the answers in the Non-Embodied/No Disruption condition, we could ensure that participants were not embodied.

Concerning the EEG measurement, we did not include this time window for the EEG analysis as participants moved their wrist at that time and their movements were different (speed, onset and direction of movement). Any conclusion we could have made regarding humans’ cognitive function on Break-in-Embodiment from this time window of EEG may be contaminated by differential movement patterns.

In page 4, line 95: “Due to time constraints, no questions were asked at the end of step 1. We ensured that participants were embodied or not during this first step thanks to the answers during Embodied/No Disruption and Non-Embodied/No Disruption condition.”

 In page 5, line 147: “We restricted the subsequent EEG analysis to the monitoring step as participants performed motor actions in the induction step, which may be a confounding factor when evaluating humans’ cognitive process.”

5. Does the duration of the embodiment phase (time interval from active induction to monitoring phase) have an impact on the results? are there differences related to the different time intervals involved?

To answer this question, we performed additional analysis to see the effect of time intervals between the two steps. As depicted in the figure below, we did not observe any difference linked to the different time intervals. To report this result, we added the following sentence in the manuscript. 

In page 8, line 191: “We did not observe any effect in EEG signals associated with different interval durations.” 

6. In general, I find the figures helpful and well thought out. In figure 1, adding labels to explain the different components of the setup could improve the quality of the picture

Thank you for this comment. We have updated figure 1 with the suggested labels.

7. Recent studies have demonstrated the relevance of the virtual reality environment (coupled with sensory stimulation) in the field of rehabilitation (Risso Preatoni et al. 2022: https://doi.org/10.1016/j.isci.2022.104129 ; Risso & Valle https://doi.org/10.1007/s40141-022-00350-x ; Serino et al. 2022 https://www.cairn.info/revue-de-neuropsychologie-2022-1-page-15.html ; Rognini et al 2019 http://dx.doi.org/10.1136/jnnp-2018-318570 ). It would be interesting to discuss the potential impact of these findings on the clinical setting.

We thank the reviewer for suggesting this interesting clinical application of our work; a sentence was added as an outlook for a clinical application, with two references. 

In page 10, line 287: “This approach could also be beneficial for the clinical evaluation of embodiment of prosthetic limbs, with benefits for reducing phantom limb distortions and pain in hand45 or leg46 amputee.”

Typos:

- Section 1 introduction : […]Errorrelated Potentials (ErrPs) induced by disruptions in VR already […]

- Section 2.2. Subjective rating: […] withingroup comparison using the Benjamini-Hochberg[…]

Thank you for pointing them out. We corrected these typos in the updated manuscript.

---

## [Decision Letter · Decision Letter 1]

28 Feb 2023

EEG signature of breaks in embodiment in VR

PONE-D-22-23485R1

Dear Dr. Herbelin,

We’re pleased to inform you that your manuscript has been judged scientifically suitable for publication and will be formally accepted for publication once it meets all outstanding technical requirements.

Kind regards,

Jane Elizabeth Aspell, PhD

Academic Editor

PLOS ONE

Additional Editor Comments (optional):

Reviewers' comments:

Reviewer's Responses to Questions

**Comments to the Author**

1. If the authors have adequately addressed your comments raised in a previous round of review and you feel that this manuscript is now acceptable for publication, you may indicate that here to bypass the “Comments to the Author” section, enter your conflict of interest statement in the “Confidential to Editor” section, and submit your "Accept" recommendation.

Reviewer #1: All comments have been addressed

Reviewer #2: All comments have been addressed

2. Is the manuscript technically sound, and do the data support the conclusions?

Reviewer #1: Yes

Reviewer #2: Yes

3. Has the statistical analysis been performed appropriately and rigorously? 

Reviewer #1: Yes

Reviewer #2: Yes

4. Have the authors made all data underlying the findings in their manuscript fully available?

Reviewer #1: Yes

Reviewer #2: (No Response)

5. Is the manuscript presented in an intelligible fashion and written in standard English?

Reviewer #1: Yes

Reviewer #2: Yes

6. Review Comments to the Author

Reviewer #1: I would like to thank the authors for addressing my comments and concerns. I find the present study highly relevant to the field and the findings can add to the already existing literature.

Reviewer #2: (No Response)

7. PLOS authors have the option to publish the peer review history of their article (what does this mean?). If published, this will include your full peer review and any attached files.

Reviewer #1: No

Reviewer #2: No

---

## [Editor Report · Acceptance letter]

9 Mar 2023

PONE-D-22-23485R1 

EEG signature of breaks in embodiment in VR 

Dear Dr. Herbelin:

I'm pleased to inform you that your manuscript has been deemed suitable for publication in PLOS ONE. Congratulations! Your manuscript is now with our production department. 

Kind regards, 

on behalf of

Dr. Jane Elizabeth Aspell 

Academic Editor

PLOS ONE